# CRISPR-FRT targets shared sites in a knock-out collection for off-the-shelf genome editing

Toon Swings [1,2], David C. Marciano[3], Benu Atri[4], Rachel E. Bosserman[5], Chen Wang [3], Marlies Leysen[1], Camille Bonte[1], Thomas Schalck [1,2], Ian Furey[6], Bram Van den Bergh [1,2], Natalie Verstraeten [1,2], Peter J. Christie[5], Christophe Herman[3], Olivier Lichtarge[3,4,6,7] & Jan Michiels [1,2]

CRISPR advances genome engineering by directing endonuclease sequence specificity with a guide RNA molecule (gRNA). For precisely targeting a gene for modification, each genetic construct requires a unique gRNA. By generating a gRNA against the flippase recognition target (FRT) site, a common genetic element shared by multiple genetic collections, CRISPR-FRT circumvents this design constraint to provide a broad platform for fast, scarless, off-the-shelf genome engineering.

[1] Centre of Microbial and Plant Genetics, KU Leuven - University of Leuven, Kasteelpark Arenberg 20, 3001 Leuven, Belgium. [2] Center for Microbiology, VIB, Kasteelpark Arenberg 20, 3001 Leuven, Belgium. [3] Department of Molecular and Human Genetics, Baylor College of Medicine, Houston, TX 77030, USA. [4] Quantitative and Computational Biosciences, Baylor College of Medicine, Houston, TX 77030, USA. [5] Department of Microbiology and Molecular Genetics, McGovern Medical School, Houston, TX 77030, USA. [6] Department of Pharmacology, Baylor College of Medicine, Houston, TX 77030, USA. [7] Computational and Integrative Biomedical Research Center, Baylor College of Medicine, Houston, TX 77030, USA. These authors contributed equally: Toon Swings, David C. Marciano. Correspondence and requests for materials should be addressed to O.L. (email: lichtarge@bcm.edu) or to J.M. (email: jan.michiels@kuleuven.vib.be)

n *E. coli*, efficient genome editing using CRISPR and homology-directed repair requires induction of the CRISPR components (Cas9 and gRNA), induction of λ phage recombinase genes[1] and a rescue DNA template with the desired mutation. Even though the mechanism of Cas9-based gene editing is still incompletely understood[2], the incorporation of a rescue template into the genome by homologous recombination likely prevents Cas9-gRNA from cutting its target sequence, thereby providing a selection wherein engineered clones survive by preventing a futile cycle of lethal double-strand DNA breakage and repair. Each engineered mutation requires a rescue template and unique gRNA that must be designed, cloned and sequence confirmed[3–6]. A recently developed CRISPR-based method (CREATE—CRISPR enabled trackable genome engineering[7]) simplifies and automates the design procedure with an algorithm that incorporates the gRNA and rescue template into a 200 nucleotide oligo that can be directly cloned into a plasmid. However, even with a 70–90% success rate of CREATE, positive clones must be screened by sequencing the modified locus. Here we make CRISPR more accessible and standardized with a simple solution that simultaneously avoids cloning of new gRNAs, circumvents complex design of rescue templates and provides an easy phenotypic screen for positive clones.

We demonstrate a method, CRISPR-FRT (Fig. 1), which directs a gRNA to a FRT (flippase recognition target) sequence present in each knockout mutant of the *E. coli* Keio collection[8]. In the arrayed Keio collection of 3884 deletion mutants, each non-essential gene of *E. coli* has been replaced by a kanamycin-resistance (KanR) cassette flanked on each end by FRT sites[7]. Instead of designing and cloning a unique gRNA for each gene, a single gRNA-FRT can target any gene that is part of the Keio collection. A Keio strain transformed with plasmids encoding the

endonuclease Cas9 and gRNA-FRT experiences a lethal double-strand DNA break at the FRT sites. Since *E. coli* naturally lacks non-homologous end joining, survival depends upon escaping a futile cycle of homology-directed repair and re-cutting by the Cas9/gRNA-FRT complex. Escape from the Cas9/gRNA-FRT complex can occur by recombination of a homologous rescue DNA template lacking a FRT site. Although any rescue DNA with homology outside the FRT-KanR-FRT cassette should be successful, in this paper we utilize mutated forms of the gene corresponding to the Keio knockout to produce specific (sometimes single nucleotide) changes to genes in their native locus. The mutated gene of interest, along with ~200–500 base pairs of homology flanking the FRT-KanR-FRT cassette, is amplified by PCR and supplied to cells with plasmids encoding gRNA-FRT and Cas9. The λ-red recombinase system[1] is also expressed in these cells to promote recombination and avoid degradation of the rescue DNA template. Recombination of the rescue DNA template results in replacement of the FRT-flanked kanamycin-resistance cassette by the gene containing the mutation of interest. Consequently, this protocol allows one to easily introduce specific mutations in the ancestral Keio background (BW25113), without any scars. Other *E. coli* strains can be engineered by first transferring the KanR cassette using P1*vir* transduction and then proceeding with the CRISPR-FRT protocol. Likewise, multiple mutations can be constructed in the same strain by consecutive cycles of P1*vir* transduction of a new gene deletion from the Keio collection followed by another round of CRISPR-FRT. Alternatively, the λ-red recombinase genes encoded on the pKDsgRNA-FRT[4] or the pCas[3] plasmid can be used to precisely replace the new target gene by a FRT-flanked KanR cassette. In this approach, a PCR-amplified oligo from the appropriate Keio clone is used as a template for homologous recombination in the

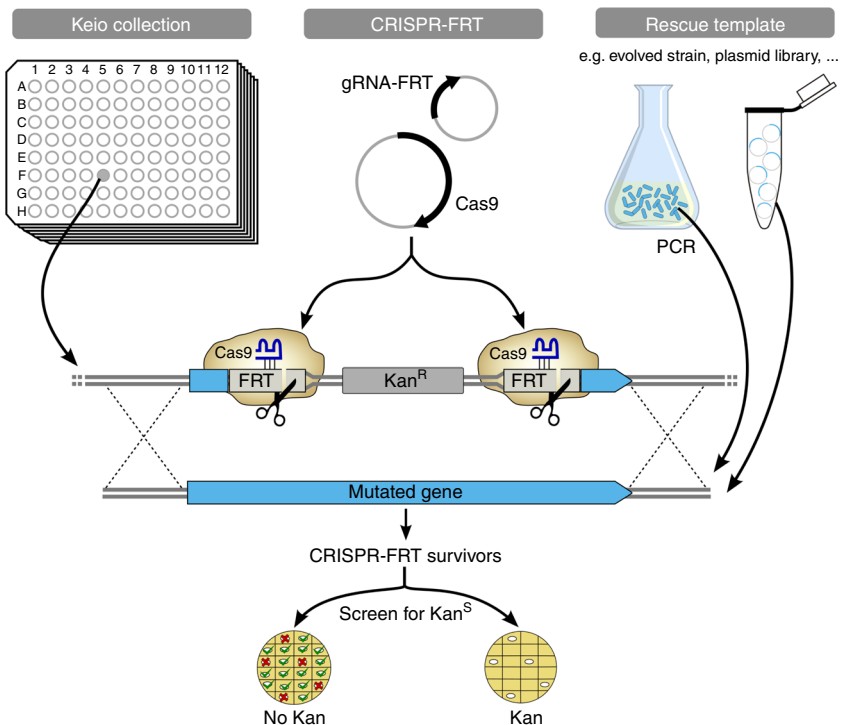

**Fig. 1** Overview of the CRISPR-FRT protocol. CRISPR-FRT makes use of the arrayed collection of Keio knockout mutants having a FRT-flanked kanamycin-resistance (KanR) cassette replacing each non-essential *E. coli* gene. CRISPR-FRT includes a gRNA-FRT that directs the Cas9 nuclease to bind and cut the two FRT sites. A convenient rescue template (e.g., a mutated gene from an evolved *E. coli* strain amplified by PCR, a plasmid-encoded gene variant, etc) recombines (dashed lines) over the homologous regions flanking the KanR cassette. Survivors are screened to separate KanR false positives (red X) from the kanamycin sensitive (KanS) true positives (green check) that replaced KanR cassette with the mutated gene

mutant background. Either approach allows for consecutive rounds of mutant construction.

## Results

**Convenience of CRISPR-FRT.** CRISPR-FRT (Fig. 1), directs a gRNA to a FRT (flippase recognition target) sequence present in each knockout mutant of the *E. coli* Keio collection[8] (Supplementary Fig. 1). There are several advantages of CRISPR-FRT. Using a single gRNA to target any non-essential *E. coli* gene obviates the need to design, clone and sequence confirm individual gRNAs for each gene or each mutation in the same gene. A static set of Cas9 and gRNA delivery plasmids can be used for each desired mutant. Also, the rescue DNA template can be easily designed. Normally, mismatches, within or adjacent to the gRNA target site are required to avoid cutting. CRISPR-FRT circumvents this aspect of rescue DNA template design by removing all traces of the FRT sites targeted by the gRNA. With this system, constructing a mutation only requires designing a set of primers to amplify the mutated gene and the corresponding Keio knockout strain transformed with the CRISPR-FRT plasmids. Furthermore, successful incorporation of the rescue DNA template results in the replacement of the $Kan^R$ cassette, thereby providing a kanamycin sensitive phenotype that can be screened. For CRISPR experiments utilizing plasmids already possessing a $Kan^R$ gene, expression of flippase can remove the $Kan^R$ cassette and leave a single FRT scar that is to be targeted by CRISPR-FRT (Supplementary Fig. 2). Without the $Kan^R$ cassette resistance phenotype, screening for true positive clones requires colony PCR. In either condition thus far, all clones producing the appropriately sized PCR DNA fragment have been confirmed by sequencing to have undergone homology-directed repair with the rescue DNA.

**Single-nucleotide editing with CRISPR-FRT.** To assess robustness and reproducibility, we cloned a gRNA targeting FRT into two independent CRISPR plasmid systems: the pTarget series[3] and the noSCAR system[4]. CRISPR-FRT was used to reconstruct mutations observed in previously conducted evolution experiments to higher ethanol tolerance[9, 10], higher persistence[11], ciprofloxacin resistance and colistin resistance in *E. coli*. In each case, rescue DNA was generated using primers designed to amplify the mutated gene with homology of 220 base pairs or more on each side of the $Kan^R$ cassette (Fig. 1). The amplified product was recombined into its native locus in either the Keio background (*E. coli* BW25113) or *E. coli* MG1655 transduced with the FRT-flanked $Kan^R$ cassette from the appropriate Keio knockout clone. The implementation of CRISPR-FRT in the pTarget system[3] first required flippase-mediated removal of the $Kan^R$ cassette to leave a single FRT target followed by introduction of the pTarget plasmids and rescue template. Clones that recombined the rescue template were then identified by colony PCR (Table 1, *acrA → yigI*). In contrast, the noSCAR[4] system plasmids were compatible with the FRT-$Kan^R$-FRT cassette and positive clones were identified by screening for kanamycin sensitivity (Table 1, *basR → mutL*). We also used CRISPR-FRT to separate several *acrR* mutants present as subpopulations within a culture of *E. coli* adapted for ciprofloxacin resistance (Fig. 2a). From a single CRISPR-FRT reaction, we recovered four unique *acrR* single substitution mutants ($M_1I$, $V_{29}$frame-shift, $K_{55}E$ and $W_{63}R$). Finally, in order to introduce mutations to two genes, we used consecutive cycles of $Kan^R$ cassette P1*vir* transduction and CRISPR-FRT of *basR* $G_{53}E$ or *basS* $C_{84}R$ into either *yejM* $P_{126}S$ or *yejM* $R_{165}C$ single mutants, respectively. Overall, we found that CRISPR-FRT functions well in either of the two CRISPR platforms tested and is adaptable to the production of multiple

mutations within or between separate genes. Moreover, CRISPR-FRT proved successful at three independent labs, demonstrating the robustness and reproducibility and making it particularly well-suited for both research and educational purposes.

In order to provide a useful demonstration of using CRISPR-FRT, we assayed several of the constructed mutants for their impact on phenotype (Fig. 3). First, the *basR* and *basS* mutants showed significantly increased colistin minimal inhibitory concentration (MIC) values and all *acrR* mutants showed significantly increased ciprofloxacin MIC values, demonstrating their effect on colistin and ciprofloxacin resistance, respectively. As expected, the mutations in *nadC* and *vacJ* which were identified as founder mutations already present in the non-resistant ancestral strain, did not result in higher ciprofloxacin MIC compared to the MIC for the wild type. Second, we showed that the single *oppB* mutation, which was previously identified in an evolution experiment to higher persistence[10], confers significantly higher persister levels both when treated with amikacin or ciprofloxacin, suggesting a putative role for this ATP-dependent oligopeptide uptake system in persistence. Next, we tested the mutation rate of mutants harboring single mutations in DNA replication and repair genes *mutL*, *mutH*, *uvrD* and *mfd*. While all mutations caused the mutation rate to increase, only the mutation rate of *mutL* ($S_{101}R$), *mutL* ($H_{270}R$), *mutH* ($W_{106}R$) and *mfd* ($V_{864}A$) was significantly increased compared to the wild type. Finally, a previously identified mutation in the *envZ* gene was assayed for ethanol tolerance[9]. Both the growth rate and final cell density increased in the *envZ* mutant compared to the wild type when exposed to a near-lethal concentration of 5% (v/v) ethanol. In general, these assays demonstrate that CRISPR-FRT allows rapid testing of different phenotypes by enabling rapid and easy introduction of multiple, separate mutations (Fig. 3).

**Targeting essential genes.** CRISPR-FRT can also be used to construct mutations in essential genes that are not part of the Keio collection. For this, the Keio knockout that is closest to the mutation in the essential gene is targeted by the gRNA-FRT. The majority of essential genes (80%) have a directly adjacent non-essential gene available in the Keio collection (Fig. 4). The rescue DNA is then extended to include homology that reaches beyond the mutation in the neighboring essential gene (Fig. 2b). Successful engineering of the essential gene thereby depends upon the homologous recombination cross-over point occurring outside the mutation in the rescue DNA. To test this feature, we used Keio knockouts of non-essential genes (*rmf* or *yejL*) to reconstruct a point mutation in neighboring, essential genes, *fabA* or *yejM* (Table 1). In each case, the wild-type non-essential gene was properly delivered while the desired mutation in the essential genes occurred in 40–44% of sequenced colonies. One of these mutations, $V_{367}A$ in *yejM*, is 1120 nucleotides from the *yejL* $Kan^R$ cassette, indicating that even relatively distant mutations can be constructed with this method. The 40–44% efficiency could be influenced by genetic distance between the mutation and the FRT site or, in specific cases, deleterious effects of mutations in the targeted essential genes. Although lower in efficiency than directly targeting a non-essential Keio knockout, the ability of CRISPR-FRT to create mutations in essential genes greatly expands its utility.

**Transferring a plasmid library to the chromosome.** In addition to being a versatile and efficient way to rapidly reconstruct single point mutations, our CRISPR-FRT method can also generate larger libraries of these point mutants in a gene's native locus (Fig. 2a). Normally, site-directed mutagenesis libraries are constructed on plasmids that can have copy number effects and

**Table 1 CRISPR-FRT efficiency across different genes and methods**

| Gene | Mutation | Rescue DNA | Survivors (CFU's/ml) | Total screened[a] | Passing screens | Sequence confirmed (%) |
|---|---|---|---|---|---|---|
| $acrA$ | $D_{268}N$ | PCR product | 9 | 4 | 2 | 100 |
| $acrR$ | multiple | PCR product | 160 | 25 | 13 | 100 |
| $deoC$ | $A_{225}V$ | PCR product | 5 | 4 | 2 | 100 |
| $malZ$ | $A_{169}V$ | PCR product | 39 | 18 | 11 | 100 |
| $nadC$ | $Y_6C$ | PCR product | 15 | 4 | 3 | 100 |
| $vacJ$ | $L_8F$ | PCR product | 38 | 4 | 3 | 100 |
| $ygiD$ | $A_{125}T$ | PCR product | 8 | 4 | 2 | 100 |
| $ygiV$ | $Q_{104}H:L_{105}V$ | PCR product | 45 | 12 | 10 | 100 |
| $yigI$ | $I_{102}T$ | PCR product | 46 | 9 | 8 | 100 |
| $basR$ | $G_{53}E$ | PCR product | 31,000 | 9 | 8 | 100 |
| $basS$ | $L_{10}P$ | PCR product | 15,000 | 9 | 8 | 100 |
| | $C_{84}R$ | PCR product | 2,500 | 9 | 8 | 100 |
| $envZ$ | $L_{116}P$ | PCR product | 48,000 | 48 | 32 | 100 |
| $acrB$ | $S_{863}A$ | PCR product | 320 | 32 | 19 | 100 |
| $fabF$ | $T_{138}A$ | PCR product | 3,000 | 3 | 2 | 100 |
| $fabR$ | $V_{92}A$ | PCR product | 13,000 | 8 | 8 | 100 |
| $oppB$ | $A_{180}E$ | PCR product | 15,000 | 40 | 10 | 100 |
| $rbsR$ | $T_{15}A$ | PCR product | 130,000 | 36 | 34 | 100 |
| $rob$ | $F_{82}L$ | PCR product | 120 | 13 | 6 | 100 |
| $mutL$ | $S_{101}R$ | PCR product | 1,160 | 10 | 10 | 100 |
| | $H_{270}R$ | PCR product | 1,240 | 10 | 7 | 100 |
| $mutH$ | $W_{108}R$ | PCR product | 200 | 10 | 10 | 100 |
| $uvrD$ | $E_{642}G$ | PCR product | 140 | 7 | 1 | 100 |
| $sspA$ | $H_{85}R$ | PCR product | 5,820 | 10 | 7 | 100 |
| | $L_{165}P$ | PCR product | 6,620 | 10 | 10 | 100 |
| | $T_{61}P$ | PCR product | 9,180 | 10 | 4 | 100 |
| | $L_{156}P$ | PCR product | 5,780 | 10 | 9 | 100 |
| $mfd$ | $V_{864}A$ | PCR product | 80 | 8 | 2 | 100 |
| $fabA$[b] | $L_{90}Q$ | PCR product | 37,000 | 20 | 5 | 40 |
| $yejM$[b] | $P_{126}S$[c] | PCR product | 3,000 | 9 | 8 | 40 |
| $yejM$[b] | $R_{165}C$[c] | PCR product | 37,000 | 9 | 9 | 44 |
| $fabF:acrB$ | $T_{138}A:S_{775}P$ | PCR product | 1,227 | 36 | 16 | 100 |
| $fabA:fabR$ | $L_{90}Q:V_{92}A$ | PCR product | 25,200 | 40 | 39 | 100 |
| $basR:yejM$ | $G_{53}E:P_{126}S$[c] | PCR product | 110,000 | 18 | 15 | 100 |
| $basS:yejM$ | $C_{84}R:R_{165}C$[c] | PCR product | 40,000 | 18 | 12 | 100 |
| $yehL$ | $S_{90}P$ | plasmid | 2,000 | 12 | 12 | 100 |
| $recA$ | multiple | plasmid | 350,000 | 100 | 77 | 100 |
| $traT$ | FLAG tag | PCR product | 240 | 24 | 9 | 100 |

[a]Clones assayed for $Kan^S$ (noSCAR system only) and yielding correct PCR fragment size
[b]Essential gene; neighboring, non-essential gene targeted
[c]$D_{347}G:G_{348}S$ $yejM$ mutations present in founder strain were also recovered

require antibiotic supplementation; an important consideration for commercial scaling of engineered strains. As a proof of principle, we transferred an existing pool of 68 $recA$ mutants from a plasmid to the $recA$ chromosomal locus. The existing $Kan^R$ $recA$ plasmid library was transformed into the Keio $recA$ knockout mutant that had been rendered kanamycin sensitive by expression of flippase and removal of the chromosomal $Kan^R$ cassette. The cells containing the library were then transformed with CRISPR-FRT plasmids and 100 out of 350,000 survivors were screened by colony PCR (Table 1). The 77 clones yielding DNA of the appropriate size were sequenced and found to have delivered the $recA$ gene from the plasmid library. The plasmid-encoded $recA$ library was collected just prior to transforming the CRISPR-FRT plasmids and sequenced to determine the library's initial sequence diversity (Fig. 5). We found no difference in diversity between the plasmid and chromosomal libraries (Multinomial goodness-of-fit test by Monte-Carlo simulation, $p$ value ± s.d. = 0.504 ± 0.002). The similar distribution of $recA$ mutants between the libraries and the large number of successful gene conversion events suggests all members of the small library were delivered to the chromosome. These results suggest other existing plasmid-based libraries could likewise be transferred to the chromosome if a homologous DNA sequence between the plasmid and target DNA locus is present.

**Engineering a protein tag into an F plasmid gene.** Finally, CRISPR-FRT can be used to engineer protein tags onto genes in their native locus. As a proof of principle, we added a FLAG-tag to the $traT$ gene in an F plasmid derivative. As no equivalent of a Keio collection exists for the F plasmid, we first replaced the $traT$ gene with the FRT-$Kan^R$-FRT cassette using an $E. coli$ strain encoding λ phage recombineering genes on the chromosome[13]. Having created an F plasmid derivative with $ΔtraT$::FRT-$Kan^R$-FRT, we next supplied the CRISPR-FRT/noSCAR plasmid set along with a $traT$-FLAG rescue DNA generated via a PCR reaction using primers designed to append the FLAG tag to the 3' end of the $traT$ gene. In contrast to editing the genome, a double-stranded DNA break caused by CRISPR-mediated cleavage of the F plasmid does not provide a direct selection for engineered clones because cells can also eliminate the target FRT sites through plasmid loss. We find plasmid loss predominates (92 of 99 clones screened) relative to the number of successful recombination events ($traT$-FLAG, 5 of 99 clones) or escape from CRISPR-FRT cutting ($ΔtraT$::$Kan^R$, 1 of 99). Inclusion of

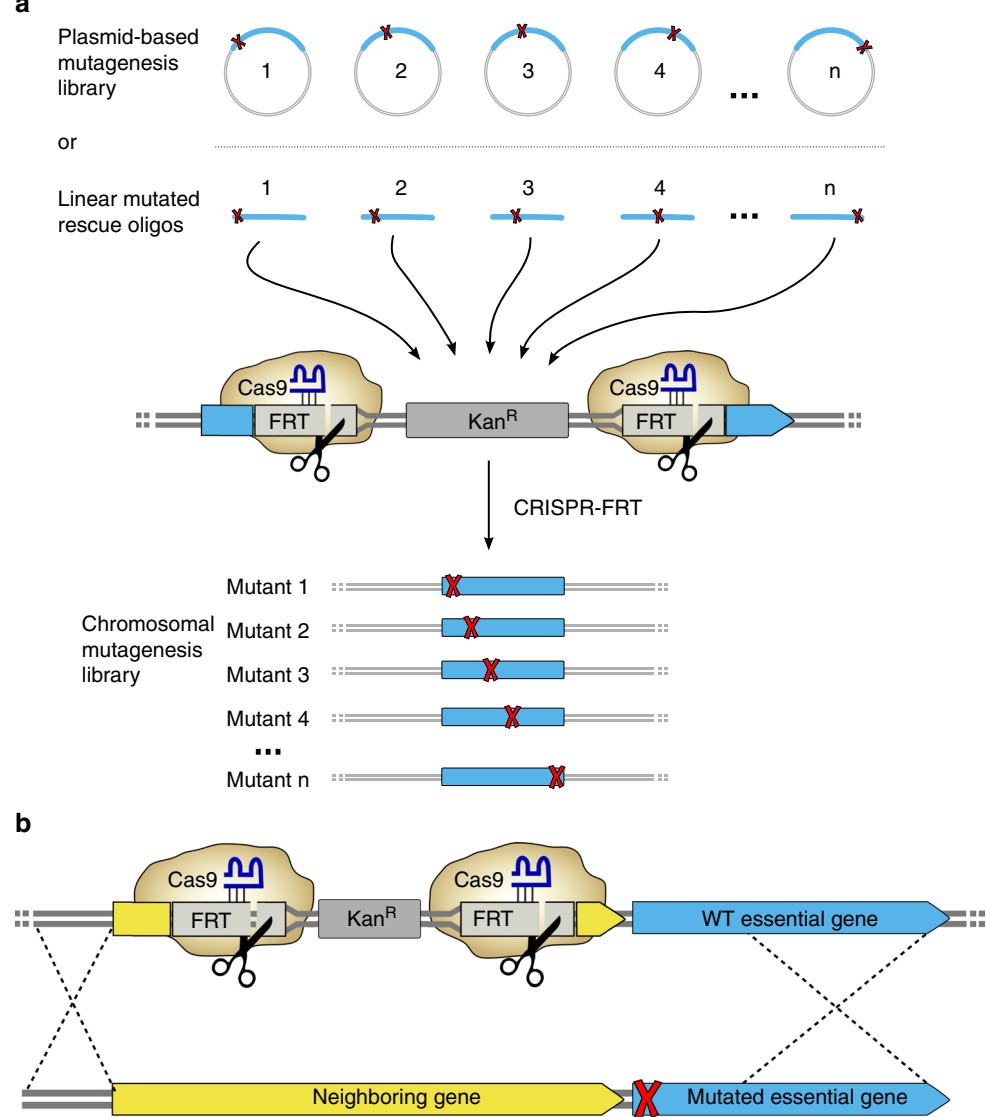

**Fig. 2** Extended applications of CRISPR-FRT. **a** Members of a plasmid-encoded mutagenesis library (1–4…*n*) or PCR-amplified linear fragments containing multiple mutated versions of one gene can serve as rescue DNA and thereby be transferred to the chromosome. This allows for single-step transfer of a mutagenesis library to the genome or reconstruction of multiple mutations (red X) in the same gene by only using a single pair of PCR primers to generate the different rescuing templates that contain the desired mutations. **b** Mutations in essential genes can be delivered to Keio strains with Kan^R cassettes inserted into the closest neighboring non-essential gene if recombination (dashed lines) occurs beyond the mutation

tetracycline in the CRISPR-FRT agar plates reduced loss of the tetracycline-resistant F plasmid derivative and improved recovery of *traT-FLAG* clones (Table 1). Although less efficient than making chromosomal point mutants, CRISPR-FRT is able to deliver FLAG-tagged *traT* to its native locus on a large plasmid. This shows the flexibility of CRISPR-FRT as a platform to engineer genes in their native locus beyond making point mutants on the chromosome.

## Discussion

CRISPR-FRT targets a single sequence that is found at a unique genetic locus for each member of a genetic library to allow for fast, robust, and scar-less genomic engineering in *E. coli*. It eliminates both the cloning of new gRNAs and the design of specific rescue DNA constructs for each desired mutation. When using current CRISPR methods, attempts to modify genes with custom gRNAs often result in unintended mutations occurring in or near the gRNA-binding site. We found that a rescue template

with the desired mutation and multiple synonymous mutations in or near the gRNA-binding site could avoid this undesired targeting of the rescue DNA. However, previous work has shown that even synonymous mutations can have phenotypic effects on their own in some cases[14, 15]. In contrast to the current methods, CRISPR-FRT permits precise single-nucleotide editing, anywhere in the gene, and avoids this specific type of off-target effect by complete ablation of the FRT target site.

This approach not only provides a ready-to-use method for reconstruction of mutations in the *E. coli* Keio collection, but is applicable to other collections with insertion sequences that can be targeted using a similar strategy. Genome library collections of 94 intergenic insertions[16] and 140 sRNA/small protein knock-outs[17] in *E. coli* and 1,052 *Salmonella enterica* gene knockouts[18] also employ a FRT-flanked Kan^R cassette that can be directly used with the CRISPR-FRT system. Gene knockout collections containing FRT sites are also available for other bacteria, including *Acinetobacter baylyi*[19] and *Burkholderia thailandensis*[20].

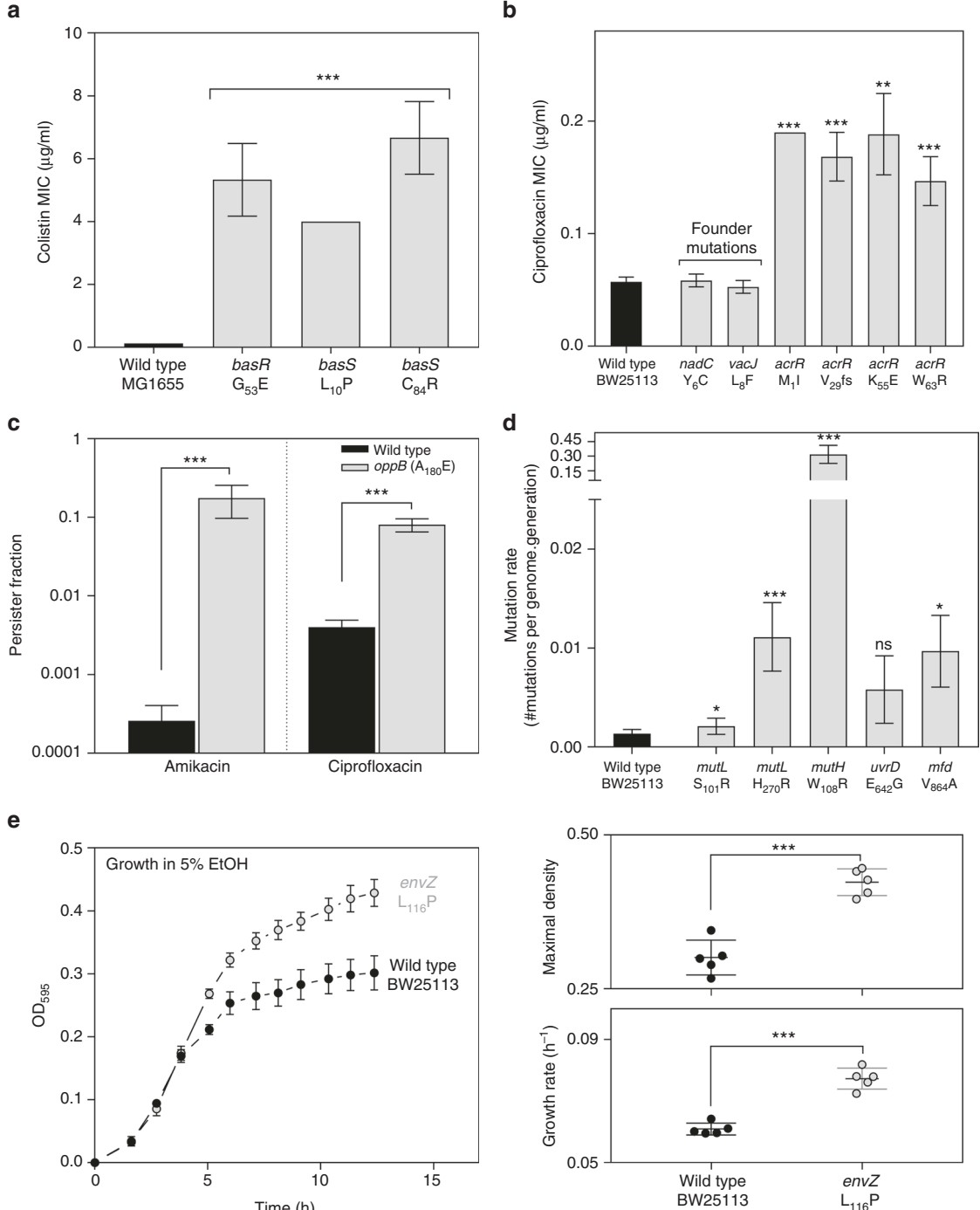

**Fig. 3** CRISPR-FRT allows rapid reconstruction and phenotypic characterization of generated mutants. **a** Mean colistin minimum inhibitory concentrations (MIC) for wild-type MG1655 and otherwise isogenic *basR* and *basS* mutants ($n = 3$; error bars represent the s.d.; two-sample unpaired t-test; ***$p < 0.001$ vs. wild type). **b** Mean ciprofloxacin minimum inhibitory concentrations (MIC) for wild-type BW25113 and otherwise isogenic *nadC*, *vacJ* and *acrR* mutants ($n \geq 3$; error bars represent standard error of the mean; two-sample unpaired t-test **$p < 0.01$, ***$p < 0.001$ vs. wild type). **c** When treated with amikacin or ciprofloxacin the *oppB* (A$_{180}$E) mutant shows a significantly increased surviving persister fraction compared to the wild type ($n = 3$; error bars represent the s.d.; two-sided t-test; ***$p < 0.001$). **d** Mutations in *mutL*, *mutH*, *uvrD* and *mfd* can change the genomic mutation rate. While all mutants show a higher mutation rate, only the *mutL* (S$_{101}$R), *mutL* (H$_{270}$R), *mutH* (W$_{106}$R) and *mfd* (V$_{864}$A) mutants show a significantly higher mutation rate compared to the wild type ($n = 24$; error bars represent upper and lower limits of the 95% confidence intervals; ShinyFlan R package built-in two-sample comparison [30]; *$p < 0.05$; ***$p < 0.001$). **e** The *envZ* (L$_{116}$P) mutant exhibits enhanced growth characteristics in the presence of 5% (v/v) ethanol compared to the wild type (left panel). Both the maximal final density (right top panel) and the growth rate (right bottom panel) significantly improved compared to the wild type ($n = 5$; error bars represent s.d.; unpaired two-sided t-test; ***$p < 0.001$). n.s. non-significant

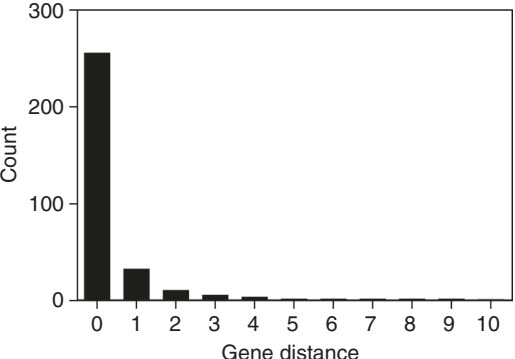

**Fig. 4** A majority of *E. coli* essential genes are adjacent to a non-essential gene that is part of the Keio knockout collection. Gene distance is a count of the minimum number of genes that separate an essential gene from a non-essential gene

Collections of gene replacements or transposon insertions, some with FRT sequences, also exist in higher organisms including *Saccharomyces cerevisiae*[21], *Drosophila melanogaster*[22], *Homo sapiens* cell lines and many more[23]. However, in organisms where non-homologous end joining strongly dominates homology-directed repair, effectiveness of CRISPR-FRT will be compromised until methods are developed to block NHEJ[24, 25]. In general, the concept of using CRISPR to target a shared sequence among insertion elements can be applied to any organism with an arrayed library and the ability to deliver a functional CRISPR system.

## Methods

**Bacterial strains and culture conditions**. To reconstruct adaptive mutations with CRISPR-FRT, we used the knockout mutants from the Keio collection derived from *E. coli* BW25113[8] or *E. coli* MG1655 that had been transduced with the FRT-Kan[R]-FRT cassette from a member of the Keio collection. These strains and all other strains and plasmids are listed in Supplementary Data 1. When the target gene was essential and no KO mutant was available, we selected the closest neighboring gene to target with CRISPR-FRT. All strains were grown in lysogeny broth (LB) medium in an orbital shaker at 200 r.p.m. and 30–32 °C or 37 °C, depending on the sensitivity of the CRISPR delivery plasmids. The HME45 strains harboring pOX38-Tc plasmid variants were grown at 30 °C.

**Plasmid and strain construction**. We obtained two plasmid sets to test the CRISPR-FRT method: the pTarget system[3] and the noSCAR system[4]. The protospacer is a 20 nt DNA stretch that defines the specificity and directs Cas9 to the position of interest. Cas9 cleaves the DNA near a PAM (-NGG) site in the DNA. Therefore, protospacers need to be designed as such that they flank a PAM site. The 48 nt-long FRT site has 4 possible PAM sites. We first used the ATUM gRNA design tool for *E. coli* K12 MG1655 (online available from https://www.atum.bio/eCommerce/cas9/input) to determine the best PAM-site with the least chance for off-target cleavage (Supplementary Fig. 1). This 20 nt sequence was used as overhangs in primers to amplify the pKDsgRNA gRNA delivery plasmid. Next, we used ligation-independent cloning or Gibson assembly (New England Biolabs, NEB) to include the FRT protospacer sequence into the pKDsgRNA gRNA delivery plasmid. This created the plasmid pKDsgRNA-FRT. For cloning the FRT-gRNA into pTargetF, we performed an inverse-PCR type reaction applied to the whole plasmid[3] using oligonucleotide primers gRNA:FRT and gRNA:LexA-G85-2 (Supplementary Data 2). However, in the latter case the resulting PCR product was cut with SpeI (NEB), the ends self-ligated with T4 DNA ligase (New England Biolabs), phenol:chloroform extracted, ethanol precipitated and transformed into DH5α cells. This created pTargetF-FRT. Due to kanamycin resistance on pCas, each Keio strain used with the pTargetF-FRT/pCas was first rendered kanamycin sensitive by flipping out the Kan[R] cassette and leaving a single FRT scar. This was achieved using *flippase* expressed from pCP20[1]. The genes targeted with the pTargetF-FRT/pCas based system included the following: *acrA*, *acrR*, *deoC*, *malZ*, *nadC*, *vacJ*, *ygiD*, *ygiV*, *yigI* and *recA*. For delivery of the pool of 68 *recA* mutants plus wild-type from a kanamycin-resistant F plasmid derivative[12], the kanamycin-resistance gene on pCas[3] was replaced by the chloramphenicol gene by recombineering. DH5α [pCas] cells were grown overnight, induced with arabinose for 3 h to express λ Red genes from pCas, made electro-competent and transformed at 1.4 kV, 25 μF, 200 Ω with overlap PCR product of the chloramphenicol gene having homology to the origin of replication, oriR101, and the tracrRNA region of the Cas9 gene. Template

for the overlap PCR included pCas, for oriR101 sequence, and pACYC184-lexA[26], for the chloramphenicol gene. Primers used in the overlap PCR reactions included repA101-1, ori101-CmR-2, ori101-Cm[R]-2-complement and Cm[R]-tracr-2-complement. Surviving clones on chloramphenicol (12.5 μg/ml) were mini-prepped, re-transformed into DH5α cells and then plated onto chloramphenicol plates again to remove unmodified pCas. The resulting plasmid is named pCas-Cm[R](+). Construction of the pool of 68 *recA* mutants on the pGE591 plasmid was facilitated by the QuikChange II XL Site-Directed Mutagenesis Kit (Agilent) and the primers listed in Supplementary Data 2. The *yehL* S[20]P rescue template originates from a genomic DNA library constructed by ligating partially Sau3AI digested DNA into BamHI digested pTargetF-FRT.

**Construction of the rescue oligo**. The rescue oligos needed to repair the double-stranded DNA breaks caused by Cas9 cleavage were constructed using PCR amplification. As a template, we used adapted strains resulting from previously run evolution experiment that harbored the specific mutation of interest[9–11]. For each mutation, we designed a primer pair that targeted a region including the entire mutated gene and 220 bp or more homology overhangs on each side (Fig. 1) (Supplementary Data 2). For the PCR reaction, we used either high-fidelity Uni-Verse polymerase (Bimake), high-fidelity Q5 polymerase (NEB) or phusion polymerase (NEB) to limit the risk for additional DNA changes during amplification. The resulting rescue oligo was purified using the DNA Clean & Concentrator™-5 (Zymo Research) and eluted in 8 μl elution buffer to obtain a highly concentrated rescue oligo (100–1000 ng/μl) for subsequent DNA transformation. For constructing *yehL* and *recA* mutants, the rescuing template was actually plasmid encoded. For the *traT*-FLAG rescue template, 499 bp upstream of the *traT* gene and the *traT* open reading frame was amplified from pOX38-Tc plasmid DNA using oligonucleotide primers orb194 and orb195. The primers were designed to incorporate a 1 × FLAG tag on the 3' end of the *traT* gene. 501 base pairs downstream of the *traT* gene were amplified from pOX38-Tc using oligonucleotide primers orb196 and orb197. The upstream and downstream PCR products were joined by fusion PCR using oligonucleotide primers orb194 and orb197. For each PCR product rescue template, the appropriately sized product was gel purified (Qiagen or Zymogen).

**CRISPR-FRT**. To reconstruct mutations from previously performed evolution experiments, we adapted two earlier described protocols[3, 4] and included the common FRT targeting protospacer in its gRNA delivery plasmid. To apply CRISPR-FRT in the two CRISPR platforms we followed instructions as previously described[3, 4]. In brief, plasmids carrying the *cas9* gene, FRT-gRNA and λ-red recombinase genes are separately transformed to the cell. Next, the recombinase genes and CRISPR genes are induced while the rescue oligo, containing the mutated gene and homologous flanking sequences, is transformed into the cell. The Cas9 nuclease will make dsDNA breaks at the FRT sites and cells only survive when the mutated genes recombined into the genome to remove the FRT sites. Resulting colonies were tested by colony PCR or streaking on a non-selective agar plate and an agar plate containing kanamycin (40 μg/ml). Colonies that showed growth on kanamycin were false positive. In most cases, the kanamycin sensitive colonies were true positives. Finally, a subset of sensitive colonies was sequenced (GATC Biotech, Germany; Genewiz, Plainfield, NJ, USA; or Baylor College of Medicine sequencing core facility, Houston, TX, USA) to confirm correct reconstruction of the adaptive mutation.

To reconstruct multiple mutations in the same background (e.g., *fabF:acrB* and *fabA:fabR*), we performed consecutive rounds of CRISPR-FRT. In case of *fabF: acrB*, after construction of the *fabF* mutation, we introduced a new FRT-flanked Kan[R] cassette to replace the native *acrB* gene. To this end, we PCR amplified the region containing the FRT-flanked Kan[R] cassette and overlapping ends on both sides. Next, the mutant harboring the first mutation was induced with arabinose (0.2%) to express the λ-red recombinase genes encoded on the pKDsgRNA-FRT[3] or the pCas9[2] plasmid. After an induction period of approximately 3 h, the PCR-amplified oligo was electroporated to the induced cells. After 1 h of incubation, transformed cells were plated on Kan plates to select for positive clones. Proper introduction of the cassette was verified by colony PCR. The positive clones were then used for a consecutive round of CRISPR-FRT to introduce the second mutation. Both Cas9 and the sgRNA delivery plasmid can be retained in the cell while performing consecutive rounds of CRISPR-FRT, shortening the time to construct multiple mutants in the same background.

**Adjacency of essential genes to non-essential genes**. CRISPR-FRT of essential genes requires the presence of a nearby Keio knockout of a non-essential gene. For the purposes of the method presented here, we define essentially as the inability to generate a clean knock-out in the Keio collection[8] without a duplication event occurring[27]. In order to determine how many essential genes lie adjacent to a non-essential Keio knockout clone, we first determined how essential genes are grouped on the *E. coli* chromosome (as singles, doubles, triples, etc.) and how many genes belonged to each class. With this information, the number of genes occurring between an essential gene and a non-essential gene was determined using a Pascal's triangle.

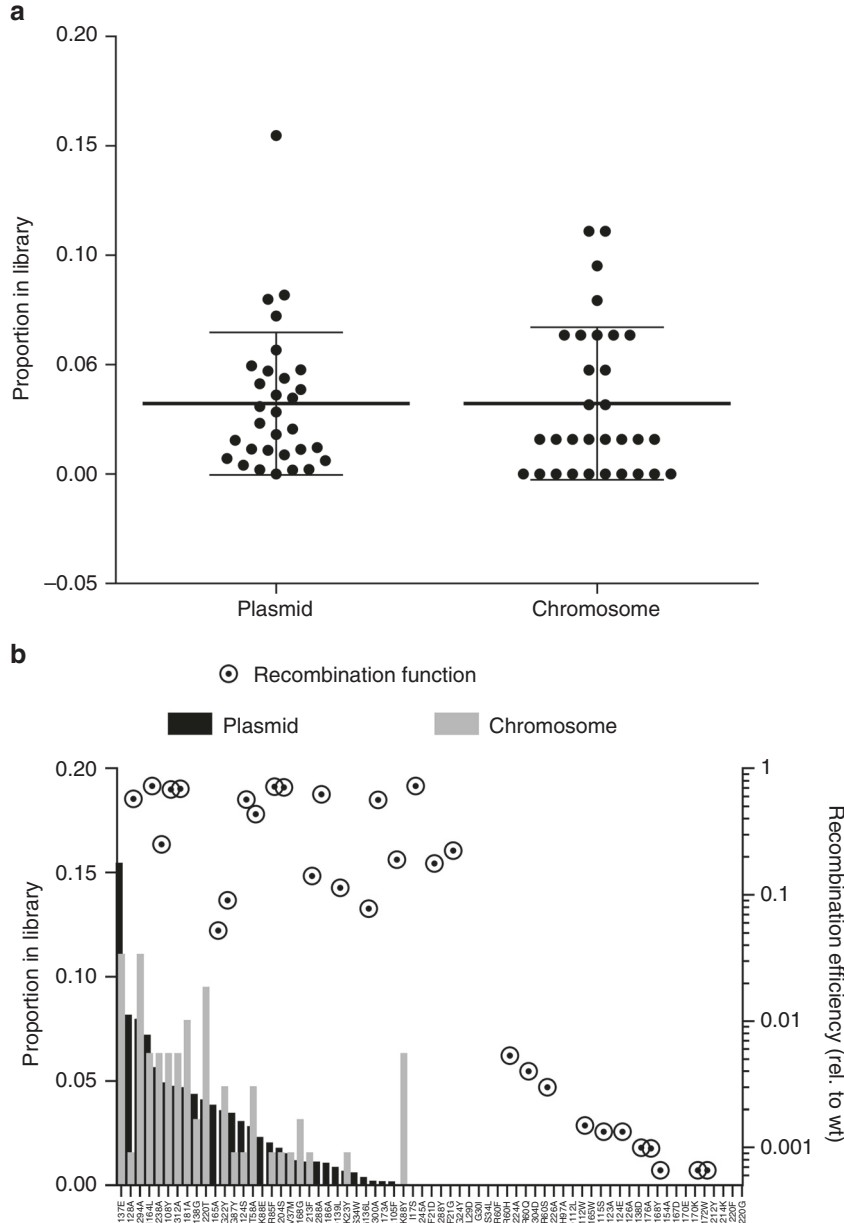

**Fig. 5** The distribution of *recA* alleles encoded by the plasmid pool before CRISPR-FRT is similar to the distribution of *recA* mutants transferred to the chromosome after CRISPR-FRT. **a** Column scatter graph showing the distribution of *recA* mutants encoded by the plasmid pool before CRISPR-FRT and *recA* mutants on the chromosome after CRISPR-FRT. Mean proportion of mutants shown with heavy bars and standard deviation shown with light bars ($n = 31$). **b** Distribution of individual *recA* mutants before and after CRISPR-FRT. Plotting previously reported[12] relative recombination function of *recA* mutants (circles, second y-axis) suggests deleterious mutants with <1% recombination activity are underrepresented in the original plasmid library pool

**CRISPR-FRT of recA library**. The small library of plasmid-encoded *recA* mutants was transferred to the chromosome using pTarget-FRT and pCas-Cm$^R$(+). To permit selection for kanamycin-resistant transformants of *recA* library plasmids, the kanamycin-resistance cassette was first removed from Δ*recA*::FRT-*Kan$^R$*-FRT Keio strain JW2669 by expression of flippase from the pCP20 and simultaneous curing of the temperature-sensitive pCP20 plasmid[1]. This generated strain DCM207 harboring the Δ*recA*::FRT scar. The small *recA* library was then transferred as a pool into DCM207. Transformants were selected on kanamycin plates, pooled, made electro-competent, transformed with pCas-Cm$^R$(+) and again selected on plates containing both chloramphenicol and kanamycin at 32 °C. The resulting colonies were again pooled, induced with arabinose to express λ-red recombinase genes from pCas-Cm$^R$(+), made electro-competent, transformed with pTarget-FRT and selected on plates containing both spectinomycin and chloramphenicol. In order to cure pTarget-FRT from these cells, the resulting colonies were pooled, diluted and plated on chloramphenicol plates containing 1 mM IPTG[3]. Individual clones were screened for replacement of the Δ*recA*::FRT scar with a *recA* gene from the plasmid pool by colony PCR (primers *mltB*-1 and

*alaS*-1, Supplementary Data 2). In order to avoid amplification of the *recA* locus from the plasmid, one primer, *mltB*-1, was designed to specifically bind the chromosome; upstream from the plasmid's cloning junction. The downstream primer, *alaS*-1, binds downstream of the *recA* gene encoded on either the plasmid or chromosome. Although not implemented here, curing of the plasmid library could be performed by directing a gRNA against a unique site in the *recA* plasmid. A new pTarget plasmid encoding an sgRNA targeting a unique sequence on the *recA* plasmid could be transformed into the chromosomal *recA* library pool and plated on chloramphenicol + spectinomycin plates to select for pCas-Cm$^R$ (+)/pTarget and cure the *recA* plasmid pool. Surviving colonies would then be pooled and again plated on IPTG + chloramphenicol to cure the new pTarget. Finally, the surviving clones would be pooled, plated onto plates without antibiotic, and incubated at 42 °C to cure pCas-Cm$^R$(+).

To determine whether the *recA* chromosomal library reflects the diversity of the plasmid library, we performed sequencing before and after conducting CRISPR-FRT. Colony PCR products from clones surviving the CRISPR reaction (see above) were Sanger sequenced (Baylor College of Medicine sequencing core) and aligned

to the *recA* gene using SnapGene v4.1.5. In order to determine the distribution of mutants in the *recA* plasmid library pool just prior to CRISPR-FRT, DCM207 [pCas-CmR(+), pGE591 library pool] cells were grown overnight, plasmids were isolated by mini-prep (Qiagen) and then used as template in a PCR reaction using primers RecA-FP and *alaS*-1. The PCR product was then prepared for deep sequencing using a Nextera XT DNA sample preparation kit and an Illumina Miseq v3, 600 cycle sequencing cartridge. The data were aligned to the *recA* sequence and SNPs were reported with their frequency using breseq software[28] with a cut off of 0.1%. A multinomial goodness-of-fit test by Monte–Carlo Simulation was performed to compare the mutation distribution in the genome with that in the plasmid using the XNomial package in R. A *p* value of 0.504 ± 0.002 was reported, suggesting that the mutations in the genome sample were a good representation of the plasmid sample.

For CRISPR-FRT of the *traT* gene on the F plasmid derivative, pOX38-Tc (gift of Laura Frost)[29], the *traT* gene was replaced with an FRT-Kan$^R$-FRT cassette from pKD13 using recombineering in *E. coli* strain HME45. *E. coli* HME45 has a defective λ prophage that retains the *gam-bet-exo* recombination genes (and others) under control of a temperature-sensitive *cI* repressor[13]. Oligonucleotide primers orb177 and orb178 were used to amplify the FRT-Kan$^R$-FRT cassette from pKD13 and provide homology for recombineering to generate pOX38-Tc-ΔtraT::FRT-Kan$^R$-FRT. The CRISPR-FRT plasmids were introduced into HME45 [pOX38-Tc-ΔtraT::FRT-Kan$^R$-FRT] and CRISPR was performed as described above with a few exceptions. Cells were grown to an OD$_{600}$ of 0.4–0.6 and then the λ-red recombinase genes encoded in the HME45 chromosome were induced by shaking in a 42 °C water bath for 15 min. As temperature induction could result in loss of the temperature-sensitive pKDsgRNA-FRT plasmid, both the *traT-FLAG* rescue DNA (810 ng) and the pKDsgRNA-FRT plasmid (370 ng) were introduced by electroporation. Cells were selected on LB agar plates supplemented with anhydrotetracycline (1000 ng/μL, Sigma), chloramphenicol, and spectinomycin and screened for plasmid retention and loss of kanamycin resistance as described above. In order to provide a selection against plasmid loss events, CRISPR-FRT was repeated as above but with the inclusion of tetracycline along with chloramphenicol, spectinomycin and anhydrotetracycline in the CRISPR/recombineering plate. Kan$^S$ colonies were screened for *traT-FLAG* using oligonucleotide primers orb179 and orb180. Three representative PCR products were verified by DNA sequencing (Macrogen, USA) with oligonucleotide primers orb179 and/or orb180.

**MIC determination**. Mutations originating from ciprofloxacin and colistin adaptive laboratory evolution experiments were tested for conferring resistance to their respective selective antibiotics by using Etest strips (bioMérieux). Overnight cultures inoculated from an isolated colony (at least three biological replicates) were diluted 100-fold into fresh lysogeny broth (LB) and allowed to grow for 2 h before being transferred to an LB plate with a sterile swab. Once the plates were dry, a single Etest strip was applied and the plate was incubated overnight at 37 °C.

**Persistence assay**. We determined the persistence fraction of wild type and *oppB* mutants in triplicate. All strains were grown for 24 h in liquid Mueller Hinton Broth (MHB) medium in an orbital shaker at 200 rpm and 37 °C. First, these cultures were diluted 100-fold in 100 mL MHB containing flasks and incubated for 16 h in an orbital shaker at 200 rpm and 37 °C. Next, the initial cell number in each flask was determined by making a dilution series and plating the $10^{-6}$ dilution on solid LB agar plates (control plates). To determine the level of persisters, 1 mL of each culture was treated with ciprofloxacin (5 μg/mL) or amikacin (400 μg/mL) for 5 h in an orbital shaker at 37 °C and 200 r.p.m. After the antibiotic treatment, samples were centrifuged for 5 min at 4.032 × *g* and the pellets were resuspended in 10 mM MgSO$_4$ to wash away the antibiotic. A 10-fold dilution series was generated starting from these resuspended cultures and dilution $10^{-2}$ and $10^{-4}$ were plated on solid LB agar plates to determine the number of surviving persister cells. Finally, the number of persister cells and the total number of cells (control plates) were used to determine the persister fraction for each tested strain[11]. Since log-transformed persister fractions are normally distributed statistical significance between the log-transformed persister fractions of wild-type and mutant cells was determined using an unpaired two-sided Student's t-test with unequal variances (based on an F-test).

**Fluctuation analysis**. We determined the genomic mutation rate of selected mutant and wild-type strains by using the Luria-Delbrück fluctuation assay[9]. In brief, overnight cultures of the strains were grown until mid-exponential phase and diluted in LB-medium to a density of 5000 cells per mL. Next, these diluted cultures were divided in at least 30 replicate cultures of 200 μL each in a 96-well plate or Eppendorf tubes and grown individually for 24 h in an orbital shaker at 200 rpm and 37 °C. After 24 h, at least 4 replicate cultures were used to determine the total cell count for each individual culture by making a 10-fold dilution series and plating on solid LB agar plates. The remaining individual cultures were entirely plated on solid LB agar plates supplemented with 100 μg/mL rifampicin to determine the number of spontaneous resistant mutants. Acquiring rifampicin resistance occurs through mutations in the *rpoB* gene. Therefore, the number of rifampicin-resistant colonies is directly related to the frequency of mutations occurring in *rpoB* and hence the mutation rate of the strain. The data were

analyzed by using flan, a recently developed R package for inference of mutation models. The software uses the number of resistant colonies in multiple individual cultures and the average number of total cells per culture to estimate the mutation rate with the Maximum Likelihood method[30]. The mutation rates of two samples were statistically compared by using the built-in two-sample test.

**Growth analysis**. To quantify the ethanol tolerance level of wild-type and mutant strains growth dynamics were monitored in the presence of 5% (v/v) ethanol. Overnight cultures were diluted 100-fold in flasks containing 50 mL LB supplemented with 5% (v/v) ethanol. The flasks were closed with rubber sealed caps to prevent ethanol evaporation. Growth of each strain was monitored in 5-fold by measuring optical density (OD$_{595nm}$) at various time points during growth. The resulting growth curves were fitted using the widely accepted Gompertz equation[31]. This fitting allowed for determination of growth rate, lag time and maximal density of wild-type and mutant strains in the presence of ethanol. Statistical significance of the difference in growth rate and maximal density was calculated using an unpaired two-sided Student's t-test with equal variances (based on an F-test).

**Data availability**. All data and plasmids are available upon request from the authors.

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

## Acknowledgements

The research was supported by the KU Leuven Research Council (PF/10/010, PDM/17/130, C1/17 3E170455), FWO (G047112N, G055517N, G0B2515N) and the VIB. The authors also gratefully acknowledge support from the National Institutes of Health (R01GM48746 to P.J.C.; R01GM088653 to C.H.; NIH-GM079656 and NIH-GM066099 to O.L.) and the National Science Foundation (NSF DBI-1356569 to O.L.). We thank Herman Dierick for helpful discussions and suggestions.

## Author contributions

T.S. designed CRISPR-FRT and its applications, performed and discussed the experiments and wrote and edited the manuscript. D.C.M. designed CRISPR-FRT and its applications, performed and discussed the experiments and wrote and edited the manuscript. B.A. generated the *recA* mutant library. R.E.B. helped in performing the experiments. C.W. helped in performing the experiments. M.L. helped in performing the experiments. C.B. helped in performing the experiments. T.S. helped in performing the experiments. B.V.d.B. helped in performing the experiments. I.F. helped in performing the experiments. N.V. discussed the results and edited the manuscript. P.J.C. discussed the results and edited the manuscript. C.H. discussed the results and edited the manuscript. O.L. discussed the results and edited the manuscript. J.M. discussed the results and edited the manuscript.

## Additional information

**Competing interests:** The authors declare no competing interests.

