## [Peer Review File · Nature Communications]

Reviewers' comments:

Reviewer #1 (Remarks to the Author):

The authors developed a CRISPR/Cas9 genome editing strategy that uses a universal FRT gRNA for *E. coli*. As the *E. coli* KEIO collection contain kanamycin resistance, by applying a screening for kanamycin sensitivity, the CRISPR-FRT strategy increases the success rate to nearly 100% for the majority of their proof-of-principle examples. Upon first glance, this work appears to have narrow applications, as it only works with organisms with available deletion collections that leave FRT scars. However, after reading the entire manuscript, I find this to be a clever system. The utility seems quite broad within organisms it does work with. I especially find the demonstration that the gRNA-FRT can work with plasmid-based libraries to be useful.

The following points should be addressed:

1. It was not clear how the *recA* mutant library plasmids were cured from the cell after CRISPR-FRT. More details are needed in the Methods section.
2. The last paragraph in the Methods section: "HME45 chromosome" needs clarification.
3. The frequency of correct gene editing for essential genes is quite low. This may be due to mutations introduced being lethal or length of rescue DNA. The authors should provide some discussion for low frequency observed.
4. Table 1 is quite confusing. How many total clones were screened to calculate % sequence confirmed? Some of the numbers under "Passing screens" was in the tens to hundreds of thousands. Did the authors actually PCR screen that many clones? And how many total were screened that did not pass?

Minor points:

1. Since the pTarget system requires the host to be kanamycin sensitive and takes away a big advantage of the CRISPR-FRT strategy to use kanamycin sensitivity to screen for false positives, the value of having it in the main text is not obvious.
2. Figure 2b and Supplementary Figure 2 are quite similar. Consider combining them.

Reviewer #2 (Remarks to the Author):

This work combines Cas9-based editing and the Keio collection of single-gene deletions in *E. coli* to streamline genome editing. An FRT-targeting sgRNA is used to direct cleavage of the inserted *kanR* cassette, and a recombineering template allows the deleted gene to be restored or modified. If the *kanR* cassette was present in the parent strain, then a simple screening step for kanamycin sensitivity can be included. The authors demonstrate the methodology by making numerous individual mutations in non-essential genes, inserting a mutation into a flanking essential gene, adding an affinity tag, and inserting a library of mutants into the same gene. In all cases, the overall editing efficiencies were as high as those reported in the previous methods for Cas9-based editing in *E. coli*.

The methodology is very clever given the simplicity of using a single sgRNA and exploiting the Keio collection, and I see it serving as a useful addition to the many techniques for Cas9-based editing in *E. coli*. What is less convincing is whether the methodology will displace these existing methods. The single sgRNA does eliminate the need for cloning, and the extra screening step based on kanamycin sensitivity does improve the probability of identifying a positive clone. However, P1 transduction is required for any strain outside of the Keio collection or when making multiple mutations that can greatly slow the process. Also, the approach relies on existing mutant genes that would have to be generated, while the traditional methods only require a different

recombineering oligo sequence. In the end, I believe the presented method is innovative and will be valuable in certain types of experiments (e.g. work with the Keio collection, introducing scarless mutations), although it will at best complement what is already available.

Other major comments:

1. The work would benefit from some useful demonstration or obtained biological insight beyond showing that the recombineering is generally efficient.
2. The ability to screen for Kan-sensitive cells is useful, although it would not be practical for any library-based approaches.
3. Parts of the manuscript read more as an advertisement rather than a scientific study (e.g. L. 151 - 157).

Other comments:

4. L. 32 - 38: the authors definitely describe the process of Cas9-based editing with the recombineering template serving to repair the DNA damage by Cas9. However, a competing explanation is that cleavage by Cas9 is irreversible, so recombineering has to occur prior to cleavage. Recent work from the Bikard group (PMID = 27060147) supports repair, although the true mechanism is far from clear.
5. L. 66: P1 transduction would be only be applicable if the genes are sufficiently far apart (e.g. >50 kb).
6. L. 107 - 108: are all essential genes immediately flanked by a non-essential gene without a large intergenic sequence? Statistics here would help support this concept.
7. L. 127 - 131: provide information on which recA mutants were identified. What was the distribution of mutants? Is this distribution reflect the diversity of the library?
8. L. 164 - 168: in most eukaryotes, NHEJ strongly dominates over HDR, which would compromise the overall effectiveness of the method in these organisms.
9. P. 10: What was the output of the ATUM gRNA design tool for each of the sites? This will help justify the selected site and address whether the other sites might also be usable.
10. Given the fact that the authors regularly recombineered into the kanR-free FRT scar, this should be depicted in the main-text or SI figures.

Response to the decision letter

Reviewer #1 (Remarks to the Author):

The authors developed a CRISPR/Cas9 genome editing strategy that uses a universal FRT gRNA for *E. coli*. As the *E. coli* Keio collection contain kanamycin resistance, by applying a screening for kanamycin sensitivity, the CRISPR-FRT strategy increases the success rate to nearly 100% for the majority of their proof-of-principle examples. Upon first glance, this work appears to have narrow applications, as it only works with organisms with available deletion collections that leave FRT scars. However, after reading the entire manuscript, I find this to be a clever system. The utility seems quite broad within organisms it does work with. I especially find the demonstration that the gRNA-FRT can work with plasmid-based libraries to be useful.

We would like to thank the reviewer for the appreciation of our work and the comments and suggestions which greatly improved our manuscript.

Reviewer #1: Major comments:

The following points should be addressed:

1. It was not clear how the *recA* mutant library plasmids were cured from the cell after CRISPR-FRT. More details are needed in the Methods section.

→ In constructing the *recA* library we made no attempt at curing the *recA* plasmid library. However, a gRNA directed against a unique sequence on the F-plasmid allows for efficient curing of the F-plasmid library. Indeed, when targeting the FRT sites on an F-based plasmid to make the *traT-FLAG* clone, we found that >90% of surviving clones had lost the plasmid if no tetracycline was provided to secure plasmid maintenance (see **lines 170-175**). In order to include details on the construction of the genomic *recA* mutant library we added the following paragraph to the methods section:

Lines 317-335: “The small library of plasmid encoded *recA* mutants was transferred to the chromosome using pTarget-FRT and pCas-Cm^R(+). To permit selection for kanamycin-resistant transformants of *recA* library plasmids, the kanamycin resistance cassette was first removed from the $\Delta recA::FRT-Kan^R-FRT$ Keio strain JW2669 by expression of flippase from the pCP20 and simultaneous curing of the temperature sensitive pCP20 plasmid, as previously described ¹. This generated strain DCM207 harboring the $\Delta recA::FRT$ scar. The small *recA* library was then transferred as a pool into DCM207. Transformants were selected on kanamycin plates, pooled, made electrocompetent, transformed with pCas-Cm^R(+) and again selected on plates containing both chloramphenicol and kanamycin at 32°C. The resulting colonies were again pooled, induced with arabinose to express λ -red recombinase genes from pCas-Cm^R(+) ², made electrocompetent, transformed with pTarget-FRT and selected on plates containing both spectinomycin and chloramphenicol. In order to cure pTarget-FRT from these cells, the resulting colonies were pooled, diluted and plated on chloramphenicol plates containing IPTG, as previously described ². Individual clones were screened for replacement of the $\Delta recA::FRT$ scar with a *recA* gene from the plasmid pool by colony PCR (primers *mltB*-1 and *alaS*-1 listed in **Supplementary Table 2**). In order to avoid amplification of the *recA* locus from the plasmid, one primer, *mltB*-1, was designed to specifically bind the chromosome; upstream from the plasmid’s cloning junction. The downstream primer, *alaS*-1, binds downstream of the *recA* gene encoded on either the plasmid or chromosome. Although not implemented here, curing of the plasmid library could be performed by directing a gRNA against a unique site in the plasmid.”

2. The last paragraph in the Methods section: "HME45 chromosome" needs clarification.

→ We clarified the origin and characteristics of the *E. coli* HME45 strain in the methods section by adding the following text:

Lines 351-352: "*E. coli* HME45 has a defective λ prophage that retains the *gam-bet-exo* recombination genes (and others) under control of a temperature sensitive *cl* repressor¹²."

3. The frequency of correct gene editing for essential genes is quite low. This may be due to mutations introduced being lethal or length of rescue DNA. The authors should provide some discussion for low frequency observed.

→ Thank you for this interesting interpretation. Indeed, the lower efficiency for gene editing in essential genes can be caused by a possible burden of these mutations as well as the length of the rescue DNA. Even though mutations in essential genes can be lethal, we did not observe any growth defects in the *fabA* (L₉₀Q) or *yejM* (P₁₂₆S and R₁₆₅C) mutants (data not shown). Moreover, the mutations that we reconstructed occurred during previous evolution experiments (Swings, *et al.*, 2017 eLife; Van den Bergh, *et al.*, 2016 Nat. Microbiol.) and were already present in strains harboring additional mutations in their genome. In this case, synthetic lethality can occur where the presence of a specific mutation is only viable when another mutation is present (Swings & Weytjens, *et al.*, 2017 Mol. Biol. Evol.). We did not observe this type of epistatic interaction as both constructed mutants were able to grow properly in the absence of other mutations. Therefore, we believe that the distance between the mutation in the essential gene and the neighboring non-essential gene is the main limiting factor for gene editing in essential genes. To address the low efficiency of gene editing in essential genes we have added the following sentence to the discussion:

Lines 141-143: "The 40-44% efficiency could be influenced by genetic distance between the mutation and the FRT site or, in specific cases, deleterious effects of mutations in the targeted essential genes."

4. Table 1 is quite confusing. How many total clones were screened to calculate % sequence confirmed? Some of the numbers under "Passing screens" was in the tens to hundreds of thousands. Did the authors actually PCR screen that many clones? And how many total were screened that did not pass?

→ We agree with the reviewer that the data in Table 1 was not entirely clear. The number "Passing screens" was calculated by taking the fraction of clones passing the screens and multiplying by the total number of surviving clones. In order to be more transparent, we have now replaced this column with two new columns showing the absolute numbers of colonies screened and the absolute number of colonies passing the screen for each constructed mutant (**Table 1**). These numbers nicely demonstrate that only a very limited number of approximately 10 colonies needs to be screened in order to find a true positive one.

Reviewer #1: Minor comments:

1. Since the pTarget system requires the host to be kanamycin sensitive and takes away a big advantage of the CRISPR-FRT strategy to use kanamycin sensitivity to screen for false positives, the value of having it in the main text is not obvious.

→ We agree with the reviewer that the kanamycin sensitivity test is a major advantage of our method. It greatly simplifies picking up true positive colonies after CRISPR editing. However, the Kan screen is not necessary as exemplified by the pCas plasmid in the pTarget system

(Jiang *et al.*, 2015 AEM) for which colonies can be easily screened by colony PCR. In fact, we constructed mutations in *acrA* → *yigL* (Table 1) as well as the *recA* chromosomal mutagenesis library with CRISPR-FRT implemented on the pTarget system. We used two CRISPR plasmid systems (i.e. noSCAR (Reisch & Prather, 2015 Sci. Rep.) and pTarget (Jiang *et al.*, 2015 AEM)) to demonstrate that CRISPR-FRT is broadly compatible with existing CRISPR systems. Therefore, we believe that is valuable to mention pTarget in the main text to show CRISPR-FRT's versatility and ability to be incorporated into future CRISPR developments.

2. Figure 2b and Supplementary Figure 2 are quite similar. Consider combining them.

→ In the revised manuscript, we have combined Figure 2b and Supplementary Figure 2 and adjusted the caption of Figure 2 accordingly (Lines 500-507).

Fig. 2: Extended applications of CRISPR-FRT. **a** Members of a plasmid-encoded mutagenesis library (1-4...n) or PCR-amplified linear fragments containing multiple mutated versions of one gene can serve as rescue DNA and thereby be transferred to the chromosome. This allows for single-step transfer of a mutagenesis library to the genome or reconstruction of multiple mutations (red X) in the same gene by only using a single pair of PCR primers to generate the different rescuing templates that contain the desired mutations. **b** Mutations in essential genes can be delivered to Keio strains with Kan^R cassettes inserted into the closest neighboring non-essential gene if recombination (dashed lines) occurs beyond the mutation.

Reviewer #2 (Remarks to the Author):

This work combines Cas9-based editing and the Keio collection of single-gene deletions in *E. coli* to streamline genome editing. An FRT-targeting sgRNA is used to direct cleavage of the inserted kan^R cassette, and a recombineering template allows the deleted gene to be restored or modified. If the kan^R cassette was present in the parent strain, then a simple screening step for kanamycin sensitivity can be included. The authors demonstrate the methodology by making numerous individual mutations in non-essential genes, inserting a mutation into a flanking essential gene, adding an affinity tag, and inserting a library of mutants into the same gene. In all cases, the overall editing efficiencies were as high as those reported in the previous methods for Cas9-based editing in *E. coli*.

→ We thank the reviewer for his or her careful consideration of our work and for the thoughtful suggestions that helped to improve our manuscript.

The methodology is very clever given the simplicity of using a single sgRNA and exploiting the Keio collection, and I see it serving as a useful addition to the many techniques for Cas9-based editing in *E. coli*. What is less convincing is whether the methodology will displace these existing methods.

→ We agree that it is difficult to predict whether or not the CRISPR-FRT method will displace current methods but this is often the case for method development. However, we would argue that CRISPR-FRT removes many levels of uncertainty including gRNA and rescue template design, and cloning of a unique gRNA. Moreover, we would like to point out that the Keio collection can be purchased for a very fair price from NBRP and that it is broadly distributed across many labs. This is also evident from >2000 paper citing the original manuscript describing its construction (Baba *et al.*, 2006 Mol. Syst. Biol.). Finally, although we utilized the Keio collection in this manuscript, we also point out other arrayed gene libraries that could be targeted by a similar approach.

The single sgRNA does eliminate the need for cloning, and the extra screening step based on kanamycin sensitivity does improve the probability of identifying a positive clone. However, P1 transduction is required for any strain outside of the Keio collection or when making multiple mutations that can greatly slow the process. Also, the approach relies on existing mutant genes that would have to be generated, while the traditional methods only require a different recombineering oligo sequence. In the end, I believe the presented method is innovative and will be valuable in certain types of experiments (e.g. work with the Keio collection, introducing scarless mutations), although it will at best complement what is already available.

→ P1 transduction is a broadly used and simple tool for genetic engineering in *E. coli* and only takes a few days to complete. However, some labs may not routinely perform this protocol. As an alternative, the existing lambda-red genes on the pKDsgRNA plasmid (or pCas-Cm^R(+)) can also be used for highly precise recombineering of the FRT-Kan^R-FRT cassette into any genetic loci of choice. This comment is also related to minor comment 5 which we address in more detail below.

→ In the manuscript we introduced mutations that occurred during a previous experimental evolution experiment by relying on existing mutant genes. However, simple overlap PCR protocols, such as the one described by Heckman and Pease (Heckman & Pease, 2007 Nat. Protocols), could be used to easily generate rescue templates containing the gene of interest mutated on any desired position. In fact, this is how we introduced the FLAG tag to *traT* on

the large, F-based plasmid pOX38-Tc (see **lines 275-281** in the Materials and Methods section). Additionally, random plasmid-based mutagenesis libraries can be generated (Wrenbeck, *et al.*, 2016 Nat. Methods) to use as template for transfer to the genome by CRISPR-FRT and screen the mutational landscape of any gene of interest.

→ We would also like to point out that existing CRISPR methods are indeed satisfactory for many experiments but the constraint of having to make extra substitutions to avoid cutting depends upon the degeneracy of codons that permits additional “silent” mutations to be introduced. Also, especially if targeting N-terminal ends of open reading frames, the consequence of introducing codon changes can have unintended consequences on protein expression levels (see Spencer *et al.* J.Mol.Bio. 2012 (ref#13) and Agashe *et al.* Mol. Biol. Evol. 2016 (ref#14). Additionally, directly targeting promoter regions would likewise benefit from scarless editing because some transcription factor binding sites may not be well defined for a given promoter and any additional substitutions could alter promoter activity.

Reviewer #2: Major comments:

1. The work would benefit from some useful demonstration or obtained biological insight beyond showing that the recombineering is generally efficient.

→ We agree that biological insight into the constructed mutations would be a very interesting addition to this manuscript. However, the main focus of the paper is to describe the method CRISPR-FRT and its broader applications. The different mutants that we reconstructed are mostly used as proof-of-concepts to demonstrate the efficiency of CRISPR-FRT. However, to address this comment we have added a new supplementary figure (**Supplementary figure 3**) that shows phenotypic data on many of the constructed mutants. We show colistin MIC values for *basR* and *basS* mutants, ciprofloxacin MIC values for *nadC*, *vacJ* and *acrR* mutants, persistence levels of the *oppB* mutant, mutation rates for *mutL*, *mutH*, *uvrD* and *mfd* mutants and ethanol tolerance data for the *envZ* mutant. These phenotypic data not so much provide extensive biological insights, but clearly demonstrate that CRISPR-FRT can be used to easily and rapidly construct several mutants that can subsequently be tested for various phenotypes.

→ We have added a new paragraph briefly describing the results of the phenotypic assays.

Lines 112-129: “... In order to provide a useful demonstration of using CRISPR-FRT, we assayed several of the constructed mutants for their impact on phenotype (**Supplementary Fig. 3**). First, the *basR* and *basS* mutants showed significantly increased colistin minimal inhibitory concentration (MIC) values and all *acrR* mutants showed significantly increased ciprofloxacin MIC values, demonstrating their effect on colistin and ciprofloxacin resistance, respectively. As expected, the mutations in *nadC* and *vacJ*, which were identified as founder mutations already present in the non-resistant ancestral strain, did not result in higher ciprofloxacin MIC compared to the MIC for the wild type. Second, we showed that the single *oppB* mutation, which was previously identified in an evolution experiment to higher persistence¹¹, confers significantly higher persister levels both when treated with amikacin or ciprofloxacin, suggesting a putative role for this ATP-dependent oligopeptide uptake system in persistence. Next, we tested the mutation rate of mutants harboring single mutations in DNA replication and repair genes *mutL*, *mutH*, *uvrD* and *mfd*. While all mutations caused the mutation rate to increase, only the mutation rate of *mutL*(H₂₇₀R) and *mutH* (W₁₀₈R) was significantly increased compared to the wild type. Finally, a previously identified mutation in the *envZ* gene was assayed for ethanol tolerance^{9,10}. Both the growth rate and final cell density increased in the *envZ* mutant compared to the wild type when exposed to a near-lethal concentration of 5% (v/v) ethanol. In general, these assays demonstrate that CRISPR-FRT allows rapid testing of different

phenotypes by enabling rapid and easy introduction of multiple, separate mutations (Supplementary Fig. 3). ...”

Supplementary Figure 3: CRISPR-FRT allows rapid reconstruction and phenotypic characterization of generated mutants. **a** Mean colistin minimum inhibitory concentrations (MIC) for wild type MG1655 and otherwise isogenic *basR* and *basS* mutants (n = 3; error bars represent the s.d., ***p<0.001 versus wild type). **b** Mean ciprofloxacin minimum inhibitory concentrations (MIC) for wild type BW25113 and otherwise isogenic *nadC*, *vacJ* and *acrR* mutants (n ≥ 3; error bars represent standard error of the mean; **p<0.01, ***p<0.001 versus wild type). **c** When treated with amikacin or ciprofloxacin the *oppB*(A₁₈₀E) mutant shows a significantly increased surviving persister fraction compared to the wild type (n = 3; error bars represent the s.d., ***p<0.001). **d** Mutations in *mutL*, *mutH*, *uvrD* and *mfd* can change the genomic mutation rate. While all mutants show a higher

mutation rate, only the *mutL*(H₂₇₀R) and *muth*(W₁₀₆R) mutants show a significantly higher mutation rate compared to the wild type (n = 24; error bars represent upper and lower limits of the 95% confidence intervals; n.s. = non-significant; ***p<0.001). **e** The *envZ*(L₁₁₆P) mutant exhibits enhanced growth characteristics in the presence of 5% (v/v) ethanol compared to the wild type (left panel). Both the maximal final density (right top panel) and the growth rate (right bottom panel) significantly improved compared to the wild type (n = 5; error bars represent s.d., ***p<0.001).

2. The ability to screen for Kan-sensitive cells is useful, although it would not be practical for any library-based approaches.

→ The ability to screen for Kan-sensitive cells provides a simple assay to identify positive clones for sequencing but it is not necessary. For instance, screening for sensitivity to kanamycin was not possible when using the pTargetF-FRT/pCas system because pCas confers kanamycin resistance. Likewise, the kanamycin sensitivity screen could not be used in identifying individual *recA* mutants because the *recA* library plasmid encodes the kanamycin resistance gene. In each case, positive clones were first screened by colony PCR prior to Sanger sequencing the PCR product. Library-based approaches would more likely depend upon selection for a specific function encoded by the gene that is being used to replace the FRT-Kan^R-FRT cassette.

3. Parts of the manuscript read more as an advertisement rather than a scientific study (e.g. L. 151 - 157).

→ We have now changed several parts of the manuscript to address this comment (e.g. **lines 109-111; lines 183-189; ...**)

Reviewer #2: Minor comments:

4. L. 32 - 38: the authors definitely describe the process of Cas9-based editing with the recombineering template serving to repair the DNA damage by Cas9. However, a competing explanation is that cleavage by Cas9 is irreversible, so recombineering has to occur prior to cleavage. Recent work from the Bikard group (PMID = 27060147) supports repair, although the true mechanism is far from clear.

→ Thank you for pointing out this interesting study by Cui and Bikard. This study aims to unravel the mechanism of Cas9 cleavage of bacterial chromosomes. Depending on the strength of the targeting, double-stranded breaks in the *E. coli* chromosomes can be repaired by homology-directed repair, which enables gene editing by specifically designing the rescue template. However, in case of strong targeting, the Cas9-induced double-strand break is irreversible, leading to cell death (Cui & Bikard, 2016 Nucl. Acids. Res.). As suggested by the reviewer, this poses the alternative explanation that homologous recombination has to occur prior to cleavage to enable genetic engineering.

→ In the revised manuscript, we have added a reference to the work by Cui and Bikard and we have edited the man text to emphasize both explanations for Cas9-based gene editing in *E. coli*.

Lines 34-38: "... Even though the mechanism of Cas9-based gene editing is still incompletely understood², the incorporation of a rescue template into the genome by homologous recombination likely prevents Cas9-gRNA from cutting its target sequence, thereby providing a selection wherein

engineered clones survive by preventing a futile cycle of lethal double-strand DNA breakage and repair. ...”

5. L. 66: P1 transduction would only be applicable if the genes are sufficiently far apart (e.g. >50 kb).

→ This is a very interesting comment. Indeed, in theory a P1 phage is able to encapsulate ~100 kb of DNA in its head. When delivering the encapsulated DNA to the donor strain the entire 100 kb region can recombine into the acceptor's genome. This poses a problem when constructing multiple consecutive mutations in genes that are less than 50 kb apart. After constructing the first mutation, a consecutive cycle of P1 transduction to replace the second gene by the FRT-flanked Kan^R cassette can possibly lead to curing of the first mutation. We have solved this issue by simply using the λ -red recombinase genes that are encoded on pKDsgRNA (Reisch & Prather, 2015 Sci. Rep.) or pCas (Jiang, *et al.*, 2015 AEM). We PCR-amplified the FRT-flanked Kan^R cassette from the appropriate Keio clone to use as a template for homologous recombination. That way we were able to precisely add new FRT-flanked Kan^R cassettes into a mutant background for consecutive cycles of mutant construction without the risk of curing previously constructed mutations. In fact, we used this method to construct the *fabF:acrB* and *fabA:fabR* double mutants (**Table 1**). We have added this to the main text and the methods section to describe this additional way of constructing multiple mutations.

Lines 69-73: "... Alternatively, the λ -red recombinase genes encoded on the pKDsgRNA-FRT³ or the pCas9² plasmid can be used to precisely replace the new targeted gene by a FRT-flanked Kan^R cassette. Therefore, a PCR-amplified oligo from the appropriate Keio clone is used as a template for homologous recombination in the mutant background. This allows for easy and precise consecutive rounds of mutant construction without the risk of curing a previously constructed one. ...”

Lines 396-307: "... To reconstruct multiple mutations in the same background (e.g. *fabF:acrB* and *fabA:fabR*), we performed consecutive rounds of CRISPR-FRT. In case of *fabF:acrB*, after construction of the *fabF* mutation, we introduced a new FRT-flanked Kan^R cassette to replace the native *acrB* gene. To this end, we PCR amplified the region containing the FRT-flanked Kan^R cassette and overlapping ends on both sides. Next, the mutant harboring the first mutation was induced with arabinose (0.2%) to express the λ -red recombinase genes encoded on the pKDsgRNA-FRT³ or the pCas9² plasmid. After an induction period of approximately 3 hours, the PCR-amplified oligo was electroporated to the induced cells. After 1 hour of incubation, transformed cells were plated on Kan plates to select for positive clones. Proper introduction of the cassette was verified by colony PCR. The positive clones were then used for a consecutive round of CRISPR-FRT to introduce the second mutation. Both Cas9 and the sgRNA delivery plasmid can be retained in the cell while performing consecutive rounds of CRISPR-FRT, shortening the time to construct multiple mutants in the same background. ...”

6. L. 107 - 108: are all essential genes immediately flanked by a non-essential gene without a large intergenic sequence? Statistics here would help support this concept.

→ This is a very good point. We examined how essential genes are distributed on the *E. coli* chromosome. The majority of essential genes (80%) have a neighboring non-essential gene (see graph below). The longest consecutive patch is 22 genes and 10.5 kb long. Fortunately, the deconvoluter library (Nehring *et al.*, Rosenberg, 2016 Nucl. Acids Res.) has an FRT-Kan^R-FRT intergenic insertion in the middle of this group and lowers the genetic distance to 5 and 5.5 kb on each side. The longest stretch of essential genes in nucleotides is actually a nontuple (9-mer) that is 11 kb long. It may be difficult to target genes in the middle of this

patch as it requires a longer PCR product (~8.1 kb) that must crossover at a point that is distal to the mutation being incorporated. In order to target these genes using CRISPR-FRT, the FRT-Kan^R-FRT cassette may need to be inserted between non-overlapping genes in this long regulon. Overall, CRISPR-FRT can target ~80% of essential genes with some additional genetic engineering being necessary to reach the remaining 20%.

We have modified the main text to help the reader understand that the majority of essential genes have a neighboring non-essential gene.

Lines 132-133: "... A majority of essential genes (80%) have a directly adjacent non-essential gene available in the Keio collection (**Supplementary Fig. 4**). ... "

→ We have also added a supplementary figure 4 that shows the distribution of consecutive essential genes:

Supplementary Figure 4: A majority of *E. coli* essential genes are adjacent to a non-essential gene that is part of the Keio knock-out collection. Gene distance is a count of the minimum number of genes that separate an essential gene from a non-essential gene.

→ We have also added a description of how this data was collected in the Materials and Methods section:

Lines 308-315: "...*Adjacency of essential genes to non-essential genes*

CRISPR-FRT of essential genes requires the presence of a nearby Keio knock-out of a non-essential gene. For the purposes of the method presented here, we define essentially as the inability to generate a clean knock-out in the Keio collection⁸ without a duplication event occurring²⁷. In order to determine how many essential genes lie adjacent to a non-essential Keio knock-out clone we first determined how essential genes are grouped on the *E. coli* chromosome (as singles, doubles, triples, etc.) and how many genes belonged to each class. With this information, the number of genes occurring between an essential gene and a non-essential gene was determined using a Pascal's triangle. ..."

7. L. 127 - 131: provide information on which *recA* mutants were identified. What was the distribution of mutants? Is this distribution reflect the diversity of the library?

→ We have identified 16 unique *recA* mutants in our initial screen of 39 clones. In order to better address this comment, we have Sanger sequenced the *recA* locus of additional clones for an overall total of 76 clones; with 21 unique *recA* mutants identified. We have also isolated plasmids from cells just prior to performing the CRISPR-FRT reaction, PCR amplified *recA*

from the plasmid and sequenced the fragment using MiSeq sequencing. Although the sequencing coverage differs, the overall distribution of *recA* mutants observed in the plasmid pool is similar to the *recA* mutants observed on the chromosome (p-value = 0.504 ± 0.002). Note that, for amplification of the chromosomal *recA* locus, we chose a primer that binds upstream of the *recA* gene and does not bind to the plasmid. In the revised manuscript, we have included a supplemental figure that shows the frequency distribution of the *recA* allele pool encoded on the plasmid (before CRISPR-FRT) and chromosome (after CRISPR-FRT). In the same supplemental figure, we have also included a panel showing the frequency of each *recA* allele observed in the plasmid pool and chromosome. Although the plasmid library and chromosomal library have similar distributions, we wondered if reduced growth rates of *recA* mutants could bias the diversity of the initial plasmid pool. In our previous publication (Adikesavan *et al.*, PLoS Genetics 2011), we determined recombination efficiencies relative to wild-type *recA* of 31 mutants in the pool. We find that mutants with <1% recombination activity constitute $\leq 0.1\%$ of the plasmid pool (our detection limit with MiSeq due to sequencing errors), suggesting a bias in the initial plasmid pool unrelated to CRISPR-FRT function. Overall, we find that CRISPR-FRT allows to deliver a plasmid library of mutants to a chromosomal locus reflective of the diversity of the input plasmid library.

→ We have updated the data in the table and edited the main text:

Lines 153-161: "... The cells containing the library were then transformed with CRISPR-FRT plasmids and 100 out of 350,000 survivors were screened by colony PCR (**Table 1**). The 77 clones yielding DNA of the appropriate size were sequenced and found to have delivered the *recA* gene from the plasmid library. The plasmid-encoded *recA* library was collected just prior to transforming the CRISPR-FRT plasmids and sequenced to determine the library's initial sequence diversity (**Supplementary Fig. 5**). We found no difference in diversity between the plasmid and chromosomal libraries (p-value 0.504 ± 0.002). The similar distribution of *recA* mutants between the libraries and the large number of successful gene conversion events suggests all members of the small library were delivered to the chromosome. ..."

→ We have also added text to the method section detailing how the sequencing was performed:

Lines 336-348: "... to determine whether the *recA* chromosomal library reflects the diversity of the plasmid library, we performed sequencing before and after conducting CRISPR-FRT. Colony PCR products from clones surviving the CRISPR reaction (see above) were Sanger sequenced (Baylor College of Medicine sequencing core) and aligned to the *recA* gene using SnapGene v4.1.5. In order to determine the distribution of mutants in the *recA* plasmid library pool just prior to CRISPR-FRT, DCM207 [pCas-Cm^R(+), pGE591 library pool] cells were grown overnight, plasmids was isolated by mini-prep (Qiagen) and then used as template in a PCR reaction using primers RecA-FP and *alaS*-1. The PCR product was then prepared for deep sequencing using a Nextera XT DNA sample preparation kit and an Illumina Miseq v3, 600 cycle sequencing cartridge. The data was aligned to the *recA* sequence and SNPs were reported with their frequency using breseq software²⁶ with a cut off of 0.1%. A multinomial goodness-of-fit test by Monte-Carlo Simulation was performed to compare the mutation distribution in the genome with that in the plasmid using the XNomial package in R. A p-value of 0.504 ± 0.002 was reported, suggesting that the mutations in the genome sample were a good representation of the plasmid sample. ... "

→ We have also added **supplementary figure 5** showing the distribution of mutations on the plasmid pool before CRISPR-FRT and on the chromosome after CRISPR-FRT:

Supplementary Figure 5: The distribution of *recA* mutants encoded by the plasmid pool before CRISPR-FRT is similar to the distribution of *recA* mutants transferred to the chromosome after CRISPR-FRT. **a** Column scatter graph showing the distribution of *recA* mutants encoded by the plasmid pool before CRISPR-FRT and *recA* mutants on the chromosome after CRISPR-FRT. Mean proportion of mutants shown with heavy bars and standard deviation shown with light bars ($n = 31$). **b** Distribution of individual *recA* mutants before and after CRISPR-FRT. Plotting previously reported²⁵ relative recombination function of *recA* mutants (circles, second y-axis) suggests deleterious mutants with <1% recombination activity are underrepresented in the original plasmid library pool.

8. L. 164 - 168: in most eukaryotes, NHEJ strongly dominates over HDR, which would compromise the overall effectiveness of the method in these organisms.

→ This is a very interesting point. Indeed, non-homologous end joining strongly dominates over homology-directed repair in most eukaryotes, thereby limiting the efficiency of all CRISPR-based gene editing approaches, including our CRISPR-FRT method. However, when blocking NHEJ, for example by targeting DNA ligase IV, a key enzyme of the NHEJ pathway using the inhibitor Scr7, the efficiency of precise genome engineering significantly increases (Maruyama, *et al.*, 2015 Nat. Biotechnol.). Moreover, recent work from the Bikard group demonstrated the ability of Csn2, a protein from the type II-A CRISPR-Cas system in *Streptococcus thermophilus*, to block NHEJ in *Bacillus subtilis* which paves the way for alternative methods to increase efficiency of precise genome editing in eukaryotes (Bernheim, *et al.*, 2017 Nat. Commun.).

→ We have added a sentence to the discussion of the manuscript to point out the effect of NHEJ.

Lines 198-200: "... However, in organisms where non-homologous end joining strongly dominates homology-directed repair, effectiveness of CRISPR-FRT will be compromised until methods are developed to block NHEJ^{23,24}. ..."

9. P. 10: What was the output of the ATUM gRNA design tool for each of the sites? This will help justify the selected site and address whether the other sites might also be usable.

→ The flippase recognition target (FRT) is a DNA stretch of only 48 nucleotides in length. Inevitably, choosing this type of short recognition sequences inherent to many libraries provides only a limited number of possible Cas9 target sites. More specifically, the FRT sequence contains 4 PAM-sites (-NGG-) at which in theory Cas9 is able to make the double-strand break. We entered the FRT sequence into the ATUM gRNA design tool and chose *Escherichia coli* K12 MG1655 as species to calculate possible off-target effects. As output of this tool determines the top 3 best gRNA which are shown in the table below. The design tool automatically assigns a score of 100 to the best predicted gRNA with the least predicted off-target effects. The other possible PAM sites can also be used, but according to the results of the design tool these gRNAs are less suitable, possibly due to more off-target effects. Therefore, we cloned the 20-nt sequence ranked highest into the gRNA delivery vector. We found this gRNA to be very effective in targeting the FRT sites as demonstrated by the efficiency of gene editing by CRISPR-FRT. Moreover, two FRT-sites that flank the Kan^R cassette are targeted simultaneously, further increasing the efficiency and specificity of the method.

Position	gRNA	Score
5-25	AGT TCCTATACTTTCTAGAGAATAGG AAC TCA AGGATATGAAAGATCTCTTATCC TTG	100
9-29	GTT CCTATACTTTCTAGAGAATAGGA ACT CAA GGATATGAAAGATCTCTTATCCT TGA	54.98
19-39	CTA GAGAATAGGAACTTCGGAATAGG AAC GAT CTCTTATCCTTGAAGCCTTATCC TTG	0.62

10. Given the fact that the authors regularly recombined into the kan^R-free FRT scar, this should be depicted in the main-text or SI figures.

→ We agree with the reviewer and to address this issue, we have added a new supplemental figure (**Supplementary Figure 2**) that depicts the CRISPR-FRT method when recombining into the Kan^R-free scar.

Supplementary Figure 2: Overview of the CRISPR-FRT protocol if targeting a single FRT site.

To use CRISPR-FRT when the Cas9 or gRNA delivery vector or the library plasmids already harbors a Kan^R cassette, first the Kan^R cassette from the Keio mutant needs to be removed by FLP recombination (see **methods**). Next, the single remaining FRT-site can be targeted by Cas9 guided by the gRNA-FRT. A convenient rescue template (e.g. a mutated gene from an evolved *E. coli* strain amplified by PCR, a plasmid-encoded gene variant, etc.) recombines (dashed lines) over the homologous regions flanking the FRT-site. Survivors are screened by colony PCR with primers flanking the FRT-site.

REVIEWERS' COMMENTS:

Reviewer #1 (Remarks to the Author):

The authors have addressed all my previous concerns.

Reviewer #2 (Remarks to the Author):

The authors sufficiently addressed all comments made by the two reviewers, which included providing phenotypic data on some of the generated mutants, determining the frequency of essential genes flanked by non-essential genes, and comparing the *recA* library in the plasmid templates and in the edited genome. They also provided a reasonable argument why the approach offers advantages over prior CRISPR methodologies in bacteria and may be widely adopted.

I have only have minor comment related to this round of revision:

The authors suggest that the *recA* plasmid could be cured by introducing a second sgRNA targeting the first plasmid. However, the new plasmid would at least need a separate resistance marker. Do the authors have something to include in the set of constructs to show that this is readily possible? And how would the new sgRNA plasmid (as well as the original sgRNA plasmid and Cas9 plasmid) be cured? While these are technical details that were not specifically explored in this, it would be important for anyone implementing the method and aiming to generate edited strains lacking the various plasmids.

Response to the decision letter

→ We thank the reviewers for their effort in reviewing and thereby improving the manuscript.

Reviewer #1:

The authors have addressed all my previous concerns.

Reviewer #2:

The authors sufficiently addressed all comments made by the two reviewers, which included providing phenotypic data on some of the generated mutants, determining the frequency of essential genes flanked by non-essential genes, and comparing the *recA* library in the plasmid templates and in the edited genome. They also provided a reasonable argument why the approach offers advantages over prior CRISPR methodologies in bacteria and may be widely adopted.

I have only have minor comment related to this round of revision:

The authors suggest that the *recA* plasmid could be cured by introducing a second sgRNA targeting the first plasmid. However, the new plasmid would at least need a separate resistance marker. Do the authors have something to include in the set of constructs to show that this is readily possible? And how would the new sgRNA plasmid (as well as the original sgRNA plasmid and Cas9 plasmid) be cured? While these are technical details that were not specifically explored in this, it would be important for anyone implementing the method and aiming to generate edited strains lacking the various plasmids.

→ We welcome reviewer #2's suggestion that we clarify precisely how one would generate edited strains that lack sgRNA, Cas9 and editing template plasmids. The general approach is to use a plasmid with a temperature sensitive origin that also encodes a sgRNA targeting the second plasmid to be cured. Once cured of the second plasmid, a temperature shift prevents replication of the first plasmid. Specifically, for delivery of the *recA* library to the chromosome, we used a derivative of the pTarget system wherein the temperature sensitive pCas-Cm^R(+) plasmid encodes a LacI repressed sgRNA targeting the pMB1 origin of replication of the second pTarget-FRT plasmid. In order to cure cells of pTarget-FRT and maintain the pCas-Cm^R(+) plasmid, the library can be plated on IPTG + chloramphenicol plates to induce expression of sgRNA-pMB1. Next, a new pTarget plasmid encoding a sgRNA targeting a unique sequence on the *recA* plasmid can be transformed and plated on chloramphenicol + spectinomycin plates to select for pCas-Cm^R(+)/pTarget and cure the *recA* plasmid pool. Surviving clones can then be pooled and again plated on IPTG + chloramphenicol to cure pTarget. Finally, the surviving clones are pooled, plated onto plates without antibiotic and incubated at 42 °C to cure pCas-Cm^R(+). All plasmids used in this study are available upon request as stated in the data availability section.

→ We added a part to the methods section in order to clarify curing op the library plasmids from edited strains:

Lines 322-328: "A new pTarget plasmid encoding an sgRNA targeting a unique sequence on the *recA* plasmid could be transformed into the chromosomal *recA* library pool and plated on chloramphenicol + spectinomycin plates to select for pCas-Cm^R(+)/pTarget and cure the *recA* plasmid pool. Surviving colonies would then be pooled and again plated on IPTG + chloramphenicol to cure the new pTarget. Finally, the surviving clones would be pooled, plated onto plates without antibiotic, and incubated at 42°C to cure pCas-Cm^R(+)."